# WikiMatrix: Mining 135M Parallel Sentences in 1620 Language Pairs from Wikipedia

## Abstract

We present an approach based on multilingual sentence embeddings to automatically extract parallel sentences from the content of Wikipedia articles in 85 languages, including several dialects or low-resource languages. We do not limit the extraction process to alignments with English, but systematically consider all possible language pairs. In total, we are able to extract 135M parallel sentences for 1620 different language pairs, out of which only 34M are aligned with English. This corpus of parallel sentences is freely available.[1]

To get an indication on the quality of the extracted bitexts, we train neural MT baseline systems on the mined data only for 1886 languages pairs, and evaluate them on the TED corpus, achieving strong BLEU scores for many language pairs. The WikiMatrix bitexts seem to be particularly interesting to train MT systems between distant languages without the need to pivot through English.

## 1 Introduction

Most of the current approaches in Natural Language Processing (NLP) are data-driven. The size of the resources used for training is often the primary concern, but the quality and a large variety of topics may be equally important. Monolingual texts are usually available in huge amounts for many topics and languages. However, multilingual resources, typically sentences in two languages which are mutual translations, are more limited, in particular when the two languages do not involve English. An important source of parallel texts are international organizations like the European Parliament (Koehn, 2005) or the United Nations (Ziemski et al., 2016). These are professional human translations, but they are in a more formal language and tend to be limited to political topics. There are several projects relying on volunteers to provide translations for public texts, e.g. news commentary (Tiedemann, 2012), OpensubTitles (Lison & Tiedemann, 2016) or the TED corpus (Qi et al., 2018)

Wikipedia is probably the largest free multilingual resource on the Internet. The content of Wikipedia is very diverse and covers many topics. Articles exist in more than 300 languages. Some content on Wikipedia was human translated from an existing article into another language, not necessarily from or into English. Eventually, the translated articles have been later independently edited and are not parallel any more. Wikipedia strongly discourages the use of unedited machine translation,[2] but the existence of such articles can not be totally excluded. Many articles have been written independently, but may nevertheless contain sentences which are mutual translations. This makes Wikipedia a very appropriate resource to mine for parallel texts for a large number of language pairs. To the best of our knowledge, this is the first work to process the entire Wikipedia and systematically mine for parallel sentences in all language pairs. We hope that this resource will be useful for several research areas and enable the development of NLP applications for more languages.

In this work, we build on a recent approach to mine parallel texts based on a distance measure in a joint multilingual sentence embedding space (Schwenk, 2018; Artetxe & Schwenk, 2018b). For this, we use the freely available LASER toolkit[3] which provides a language agnostic sentence encoder which was trained on 93 languages (Artetxe & Schwenk, 2018a). We approach the computational

---

[1] Anonymized for review
[2] https://en.wikipedia.org/wiki/Wikipedia:Translation
[3] https://github.com/facebookresearch/LASER

challenge to mine in almost six hundred million sentences by using fast indexing and similarity search algorithms.

The paper is organized as follows. In the next section, we first discuss related work. We then summarize the underlying mining approach. Section 4 describes in detail how we applied this approach to extract parallel sentences from Wikipedia in 1620 language pairs. To asses the quality of the extracted bitexts, we train NMT systems for a subset of language pairs and evaluate them on the TED corpus (Qi et al., 2018) for 45 languages. These results are presented in section 5. The paper concludes with a discussion of future research directions.

## 2 RELATED WORK

There is a large body of research on mining parallel sentences in collections of monolingual texts, usually named *"comparable coprora"*. Initial approaches to bitext mining have relied on heavily engineered systems often based on metadata information, e.g. (Resnik, 1999; Resnik & Smith, 2003). More recent methods explore the textual content of the comparable documents. For instance, it was proposed to rely on cross-lingual document retrieval, e.g. (Utiyama & Isahara, 2003; Munteanu & Marcu, 2005) or machine translation, e.g. (Abdul-Rauf & Schwenk, 2009; Bouamor & Sajjad, 2018), typically to obtain an initial alignment that is then further filtered. In the shared task for bilingual document alignment (Buck & Koehn, 2016), many participants used techniques based on n-gram or neural language models, neural translation models and bag-of-words lexical translation probabilities for scoring candidate document pairs. The STACC method uses seed lexical translations induced from IBM alignments, which are combined with set expansion operations to score translation candidates through the Jaccard similarity coefficient (Etchegoyhen & Azpeitia, 2016; Azpeitia et al., 2017; 2018). Using multilingual noisy web-crawls such as ParaCrawl[4] for filtering good quality sentence pairs has been explored in the shared tasks for high resource (Koehn et al., 2018) and low resource (Koehn et al., 2019) languages.

In this work, we rely on massively multilingual sentence embeddings and margin-based mining in the joint embedding space, as described in (Schwenk, 2018; Artetxe & Schwenk, 2018b;a). This approach has also proven to perform best in a low resource scenario (Chaudhary et al., 2019; Koehn et al., 2019). Closest to this approach is the research described in España-Bonet et al. (2017); Hassan et al. (2018); Guo et al. (2018); Yang et al. (2019). However, in all these works, only bilingual sentence representations have been trained. Such an approach does not scale to many languages, in particular when considering all possible language pairs in Wikipedia. Finally, related ideas have been also proposed in Bouamor & Sajjad (2018) or Grégoire & Langlais (2017). However, in those works, mining is not solely based on multilingual sentence embeddings, but they are part of a larger system. To the best of our knowledge, this work is the first one that applies the same mining approach to all combinations of many different languages, written in more than twenty different scripts.

Wikipedia is arguably the largest comparable corpus. One of the first attempts to exploit this resource was performed by Adafre & de Rijke (2006). An MT system was used to translate Dutch sentences into English and to compare them with the English texts. This method yielded several hundreds of Dutch/English parallel sentences. Later, a similar technique was applied to the Persian/English pair (Mohammadi & GhasemAghaee, 2010). Structural information in Wikipedia such as the topic categories of documents was used in the alignment of multilingual corpora (Otero & López, 2010). In another work, the mining approach of Munteanu & Marcu (2005) was applied to extract large corpora from Wikipedia in sixteen languages (Smith et al., 2010). Otero et al. (2011) measured the comparability of Wikipedia corpora by the translation equivalents on three languages Portuguese, Spanish, and English. Patry & Langlais (2011) came up with a set of features such as Wikipedia entities to recognize parallel documents, and their approach was limited to a bilingual setting. Tufis et al. (2013) proposed an approach to mine parallel sentences from Wikipedia textual content, but they only considered high-resource languages, namely German, Spanish and Romanian paired with English. Tsai & Roth (2016) grounded multilingual mentions to English wikipedia by training cross-lingual embeddings on twelve languages. Gottschalk & Demidova (2017) searched for parallel text passages in Wikipedia by comparing their named entities and time expressions. Finally, Aghaebrahimian (2018) propose an approach based on bilingual BiLSTM sentence encoders to mine German, French and Persian parallel texts with English. Parallel data consisting of aligned

---

[4]http://www.paracrawl.eu/

Wikipedia titles have been extracted for twenty-three languages[5]. Since Wikipedia titles are rarely entire sentences with a subject, verb and object, it seems that only modest improvements were observed when adding this resource to the training material of NMT systems.

We are not aware of other attempts to systematically mine for parallel sentences in the textual content of Wikipedia for a large number of languages.

## 3    DISTANCE-BASED MINING APPROACH

The underling idea of the mining approach used in this work is to first learn a multilingual sentence embedding, i.e. an embedding space in which semantically similar sentences are close independently of the language they are written in. This means that the distance in that space can be used as an indicator whether two sentences are mutual translations or not. Using a simple absolute threshold on the cosine distance was shown to achieve competitive results (Schwenk, 2018). However, it has been observed that an absolute threshold on the cosine distance is globally not consistent, e.g. (Guo et al., 2018). The difficulty to select one global threshold is emphasized in our setting since we are mining parallel sentences for many different language pairs.

### 3.1    MARGIN CRITERION

The alignment quality can be substantially improved by using a margin criterion instead of an absolute threshold (Artetxe & Schwenk, 2018b). In that work, the margin between two candidate sentences $x$ and $y$ is defined as the ratio between the cosine distance between the two sentence embeddings, and the average cosine similarity of its nearest neighbors in both directions:

$$\text{margin}(x, y) = \frac{\cos(x, y)}{\displaystyle\sum_{z \in \text{NN}_k(x)} \frac{\cos(x, z)}{2k} + \sum_{z \in \text{NN}_k(y)} \frac{\cos(y, z)}{2k}} \tag{1}$$

where $\text{NN}_k(x)$ denotes the $k$ unique nearest neighbors of $x$ in the other language, and analogously for $\text{NN}_k(y)$. We used $k = 4$ in all experiments.

We follow the *"max"* strategy as described in (Artetxe & Schwenk, 2018b): the margin is first calculated in both directions for all sentences in language $L_1$ and $L_2$. We then create the union of these forward and backward candidates. Candidates are sorted and pairs with source or target sentences which were already used are omitted. We then apply a threshold on the margin score to decide whether two sentences are mutual translations or not. Note that with this technique, we always get the same aligned sentences, independently of the mining direction, e.g. searching translations of French sentences in a German corpus, or in the opposite direction. The reader is referred to Artetxe & Schwenk (2018b) for a detailed discussion with related work.

The complexity of a distance-based mining approach is $O(N \times M)$, where $N$ and $M$ are the number of sentences in each monolingual corpus. This makes a brute-force approach with exhaustive distance calculations intractable for large corpora. Margin-based mining was shown to significantly outperform the state-of-the-art on the shared-task of the workshop on Building and Using Comparable Corpora (BUCC) (Artetxe & Schwenk, 2018b). The corpora in the BUCC corpus are rather small: at most 567k sentences.

The languages with the largest Wikipedia are English and German with 134M and 51M sentences, respectively, after pre-processing (see Section 4.1 for details). This would require $6.8 \times 10^{15}$ distance calculations.[6] We show in Section 3.3 how to tackle this computational challenge.

### 3.2    MULTILINGUAL SENTENCE EMBEDDINGS

Distance-based bitext mining requires a joint sentence embedding for all the considered languages. One may be tempted to train a bi-lingual embedding for each language pair, e.g. (España-Bonet

---

[5]https://linguatools.org/tools/corpora/wikipedia-parallel-titles-corpora/
[6]Strictly speaking, Cebuano and Swedish are larger than German, yet mostly consist of template/machine translated text https://en.wikipedia.org/wiki/List_of_Wikipedias

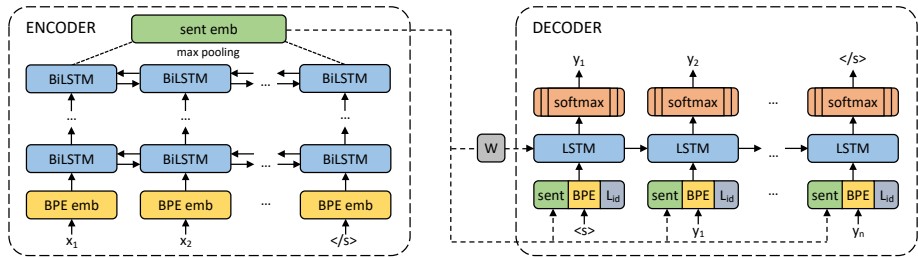

Table 1: Architecture of the system used to train massively multilingual sentence embeddings. See Artetxe & Schwenk (2018a) for details.

et al., 2017; Hassan et al., 2018; Guo et al., 2018; Yang et al., 2019), but this is difficult to scale to thousands of language pairs present in Wikipedia. Instead, we chose to use one single massively multilingual sentence embedding for all languages, namely the one proposed by the open-source LASER toolkit (Artetxe & Schwenk, 2018a). Training one joint multilingual embedding on many languages at once also has the advantage that low-resource languages can benefit from the similarity to other language in the same language family. For example, we were able to mine parallel data for several Romance (minority) languages like Aragonese, Lombard, Mirandese or Sicilian although data in those languages was not used to train the multilingual LASER embeddings.

The underlying idea of LASER is to train a sequence-to-sequence system on many language pairs at once using a shared BPE vocabulary and a shared encoder for all languages. The sentence representation is obtained by max-pooling over all encoder output states. Figure 1 illustrates this approach. The reader is referred to Artetxe & Schwenk (2018a) for a detailed description.

### 3.3 FAST SIMILARITY SEARCH

Fast large-scale similarity search is an area with a large body of research. Traditionally, the application domain is image search, but the algorithms are generic and can be applied to any type of vectors. In this work, we use the open-source FAISS library[7] which implements highly efficient algorithms to perform similarity search on billions of vectors (Johnson et al., 2017). An additional advantage is that FAISS has support to run on multiple GPUs. Our sentence representations are 1024-dimensional. This means that the embeddings of all English sentences require $153 \cdot 10^6 \times 1024 \times 4 = 513$ GB of memory. Therefore, dimensionality reduction and data compression are needed for efficient search. In this work, we chose a rather aggressive compression based on a 64-bit product-quantizer (Jégou et al., 2011), and portioning the search space in 32k cells. This corresponds to the index type "`OPQ64,IVF32768,PQ64`" in FAISS terms.[8] Another interesting compression method is scalar quantization. A detailed comparison is left for future research. We build and train one FAISS index for each language.

The compressed FAISS index for English requires only 9.2GB, i.e. more than fifty times smaller than the original sentences embeddings. This makes it possible to load the whole index on a standard GPU and to run the search in a very efficient way on multiple GPUs in parallel, without the need to shard the index. The overall mining process for German/English requires less than 3.5 hours on 8 GPUs, including the nearest neighbor search in both direction and scoring all candidates

## 4 BITEXT MINING IN WIKIPEDIA

For each Wikipedia article, it is possible to get the link to the corresponding article in other languages. This could be used to mine sentences limited to the respective articles. One one hand, this **local mining** has several advantages: 1) mining is very fast since each article usually has a few hundreds of sentences only; 2) it seems reasonable to assume that a translation of a sentence is more likely to be found in the same article than anywhere in the whole Wikipedia. On the other hand, we

---

[7]https://github.com/facebookresearch/faiss
[8]https://github.com/facebookresearch/faiss/wiki/Faiss-indexes

| $L_1$ (French) | *Ceci est une très grande maison* |
| --- | --- |
| $L_2$ (German) | *Das ist ein sehr großes Haus* |
| | *This is a very big house* |
| | *Ez egy nagyon nagy ház* |
| | *Ini rumah yang sangat besar* |

Table 2: Illustration how sentences in the wrong language can hurt the alignment process with a margin criterion. See text for a detailed discussion.

hypothesize that the margin criterion will be less efficient since one article has usually few sentences which are similar. This may lead to many sentences in the overall mined corpus of the type *"NAME was born on DATE in CITY"*, *"BUILDING is a monument in CITY built on DATE"*, etc. Although those alignments may be correct, we hypothesize that they are of limited use to train an NMT system, in particular when they are too frequent. In general, there is a risk that we will get sentences which are close in structure and content.

The other option is to consider the whole Wikipedia for each language: for each sentence in the source language, we mine in all target sentences. This **global mining** has several potential advantages: 1) we can try to align two languages even though there are only few articles in common; 2) many short sentences which only differ by the name entities are likely to be excluded by the margin criterion. A drawback of this global mining is a potentially increased risk of misalignment and a lower recall.

In this work, we chose the global mining option. This will allow us to scale the same approach to other, potentially huge, corpora for which document-level alignments are not easily available, e.g. Common Crawl. An in depth comparison of local and global mining (on Wikipedia) is left for future research.

### 4.1 CORPUS PREPARATION

Extracting the textual content of Wikipedia articles in all languages is a rather challenging task, i.e. removing all tables, pictures, citations, footnotes or formatting markup. There are several ways to download Wikipedia content. In this study, we use the so-called *CirrusSearch dumps* since they directly provide the textual content without any meta information.[9] We downloaded this dump in March 2019. A total of about 300 languages are available, but the size obviously varies a lot between languages. We applied the following processing:

- extract the textual content;
- split the paragraphs into sentences;
- remove duplicate sentences;
- perform language identification and remove sentences which are not in the expected language (usually, citations or references to texts in another language).

It should be pointed out that sentence segmentation is not a trivial task, with many exceptions and specific rules for the various languages. For instance, it is rather difficult to make an exhaustive list of common abbreviations for all languages. In German, points are used after numbers in enumerations, but numbers may also appear at the end of sentences. Other languages do not use specific symbols to mark the end of a sentence, namely Thai. We are not aware of a reliable and freely available sentence segmenter for Thai and we had to exclude that language. We used the freely available Python tool[10] which is based on Moses scripts. Regular expressions were used for most of the Asian languages, falling back to English for the remaining languages. This gives us 879 million sentences in 300 languages. The margin criterion to mine for parallel data requires that the texts do not contain duplicates. This removes about 25% of the sentences.[11]

---

[9] https://dumps.wikimedia.org/other/cirrussearch/
[10] https://pypi.org/project/sentence-splitter/
[11] The Cebuano and Waray Wikipedia were largely created by a bot and contain more than 65% of duplicates.

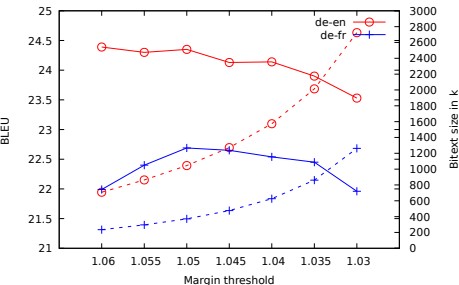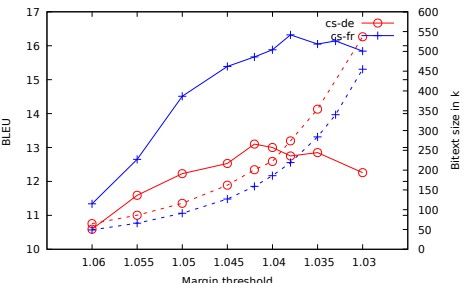

Figure 1: BLEU scores (continuous lines) for several NMT systems trained on bitexts extracted from Wikipedia for different margin thresholds. The size of the mined bitexts are depicted as dashed lines.

LASER's sentence embeddings are totally language agnostic. This has the side effect that the sentences in other languages (e.g. citations or quotes) may be considered closer in the embedding space than a potential translation in the target language. Table 2 illustrates this problem. The algorithm would not select the German sentence although it is a perfect translation. The sentences in the other languages are also valid translations which would yield a very small margin. To avoid this problem, we perform language identification (LID) on all sentences and remove those which are not in the expected language. LID is performed with fasttext[12] (Joulin et al., 2016). Fasttext does not support all the 300 languages present in Wikipedia and we disregarded the missing ones (which typically have only few sentences anyway). After deduplication and LID, we dispose of 595M sentences in 182 languages. English accounts for 134M sentences, and German with 51M sentences is the second largest language. The sizes for all languages are given in Tables 4 and 6.

## 4.2 THRESHOLD OPTIMIZATION

Artetxe & Schwenk (2018b) optimized their mining approach for each language pair on a provided corpus of gold alignments. This is not possible when mining Wikipedia, in particular when considering many language pairs. In this work, we use an evaluation protocol inspired by the WMT shared task on parallel corpus filtering for low-resource conditions (Koehn et al., 2019): an NMT system is trained on the extracted bitexts – for different thresholds – and the resulting BLEU scores are compared. We choose newstest2014 of the WMT evaluations since it provides an $N$-way parallel test sets for English, French, German and Czech. We favoured the translation between two morphologically rich languages from different families and considered the following language pairs: German/English, German/French, Czech/German and Czech/French. The size of mined bitexts is in the range of 100k to more than 2M (see Table 3 and Figure 1). We did not try to optimize the architecture of the NMT system to the size of the bitexts and used the same architecture for all systems: the encoder and decoder are 5-layer transformer models as implemented in fairseq (Ott et al., 2019). The goal of this study is not to develop the best performing NMT system for the considered languages pairs, but to compare different mining parameters.

The evolution of the BLEU score in function of the margin threshold is given in Figure 1. Decreasing the threshold naturally leads to more mined data – we observe an exponential increase of the data size. The performance of the NMT systems trained on the mined data seems to change as expected, in a surprisingly smooth way. The BLEU score first improves with increasing amounts of available training data, reaches a maximum and than decreases since the additional data gets more and more noisy, i.e. contains wrong translations. It is also not surprising that a careful choice of the margin threshold is more important in a low-resource setting. Every additional parallel sentence is important. According to Figure 1, the optimal value of the margin threshold seems to be 1.05 when many sentences can be extracted, in our case German/English and German/French. When less parallel data is available, i.e. Czech/German and Czech/French, a value in the range of 1.03–1.04 seems to be a better choice. Aiming at one threshold for all language pairs, we chose a value of 1.04. It seems to be a good compromise for most language pairs. However, for the open release of this corpus, we provide all mined sentence with a margin of 1.02 or better. This would enable end users

---

[12]https://fasttext.cc/docs/en/language-identification.html

| Bitexts | de-en | de-fr | cs-de | cs-fr |
|---|---|---|---|---|
| Europarl | 1.9M 21.5 | 1.9M 23.6 | 568k 14.9 | 627k 21.5 |
|  | 1.0M 21.2 | 370k 21.1 | 200k 12.6 | 220k 19.2 |
| Mined Wikipedia | 1.0M 24.4 | 372k 22.7 | 201k 13.1 | 219k 16.3 |
| Europarl + Wikipedia | 3.0M 25.5 | 2.3M 25.6 | 768k 17.7 | 846k 24.0 |

Table 3: Comparison of NMT systems trained on the Europarl corpus and on bitexts automatically mined in Wikipedia by our approach at a threshold of 1.04. We give the number of sentences (first line) and the BLEU score (second line of each bloc) on `newstest2014`.

to choose an optimal threshold for their particular applications. However, it should be emphasized that we do not expect that many sentence pairs with a margin as low as 1.02 are good translations.

For comparison, we also trained NMT systems on the Europarl corpus V7 (Koehn, 2005), i.e. professional human translations, first on all available data, and then on the same number of sentences than the mined ones (see Table 3). With the exception of Czech/French, we were able to achieve better BLEU scores with the automatically mined bitexts in Wikipedia than with Europarl of the same size. Adding the mined text to the full Europarl corpus, also leads to further improvements of 1.1 to 3.1 BLEU. We argue that this is a good indicator of the quality of the automatically extracted parallel sentences.

## 5 RESULT ANALYSIS

We run the alignment process for all possible combinations of languages in Wikipedia. This yielded 1620 language pairs for which we were able to mine at least ten thousand sentences. Remember that mining $L_1 \rightarrow L_2$ is identical to $L_2 \rightarrow L_1$, and is counted only once. We propose to analyze and evaluate the extracted bitexts in two ways. First, we discuss the amount of extracted sentences (Section 5.1). We then turn to a qualitative assessment by training NMT systems for all language pairs with more than twenty-five thousand mined sentences (Section 5.2).

### 5.1 QUANTITATIVE ANALYSIS

Due to space limits, Table 4 summarizes the number of extracted parallel sentences only for languages which have a total of at least five hundred thousand parallel sentences (with all other languages at a margin threshold of 1.04). Additional results are given in Table 6 in the Appendix.

There are many reasons which can influence the number of mined sentences. Obviously, the larger the monolingual texts, the more likely it is to mine many parallel sentences. Not surprisingly, we observe that more sentences could be mined when English is one of the two languages. Let us point out some languages for which it is usually not obvious to find parallel data with English, namely Indonesian (1M), Hebrew (545k), Farsi (303k) or Marathi (124k sentences). The largest mined texts not involving English are Russian/Ukrainian (2.5M), Catalan/Spanish (1.6M), between the Romance languages French, Spanish, Italian and Portuguese (480k–923k), and German/French (626k).

It is striking to see that we were able to mine more sentences when Galician and Catalan are paired with Spanish than with English. On one hand, this could be explained by the fact that LASER's multilingual sentence embeddings may be better since the involved languages are linguistically very similar. On the other, it could be that the Wikipedia articles in both languages share a lot of content, or are obtained by mutual translation.

Services from the European Commission provide human translations of (legal) texts in all the 24 official languages of the European Union. This N-way parallel corpus enables training of MT system to directly translate between these languages, without the need to pivot through English. This is

Table 4: WikiMatrix: number of extracted sentences for each language pair (in thousands), e.g. Es-Ja=219 corresponds to 219,260 sentences (rounded). The column "*size*" gives the number of lines in the monolingual texts after deduplication and LID.

| Src/Trg | ar | bg | bs | cs | da | de | el | en | eo | es | fr | fr-ca | gl | he | hr | hu | id | it | ja | ko | mk | nb | nl | pl | pt | pt-br | ro | ru | sk | sl | sr | sv | tr | uk | vi | zh-cn | zh-tw |
|---|---|---|---|---|---|---|---|---|---|---|---|---|---|---|---|---|---|---|---|---|---|---|---|---|---|---|---|---|---|---|---|---|---|---|---|---|---|
| ar | | 4.9 | 1.8 | 3.0 | 4.5 | 3.8 | 6.7 | 20.3 | 4.1 | 13.2 | 12.2 | 9.0 | 5.6 | 3.5 | 2.2 | 2.7 | 9.2 | 9.9 | 4.2 | 5.3 | 5.5 | 4.9 | 4.4 | 3.0 | 12.0 | 12.2 | 5.6 | 5.6 | 1.5 | 2.7 | 1.2 | 4.0 | 2.4 | 4.5 | 12.3 | 8.2 | 4.9 |
| bg | 3.0 | | 4.2 | 6.7 | 8.7 | 8.5 | 10.2 | 25.3 | 7.7 | 16.3 | 14.7 | 11.9 | 8.4 | 3.2 | 6.4 | 4.7 | 10.3 | 14.7 | 11.9 | 4.0 | 5.4 | 16.2 | 8.1 | 7.7 | 6.3 | 14.7 | 15.0 | 9.8 | 12.4 | 4.0 | 6.4 | 2.7 | 7.1 | 2.4 | 10.4 | 11.7 | 6.4 |
| bs | 1.2 | 6.1 | | 4.1 | 3.7 | 5.8 | 4.5 | 21.7 | | 9.9 | 6.7 | 4.6 | 5.7 | 1.0 | 30.9 | 2.4 | 5.3 | 6.5 | 1.4 | | 12.9 | | 3.8 | 2.9 | 9.9 | 10.9 | 5.7 | 4.8 | 3.1 | 10.4 | 10.6 | 4.4 | 1.5 | 5.4 | 5.8 | 2.8 | 1.8 |
| cs | 1.9 | 7.8 | 3.7 | | 7.1 | 8.3 | 6.4 | 20.0 | 10.4 | 12.6 | 11.4 | 8.6 | 5.0 | 2.5 | 6.5 | 4.9 | 7.8 | 9.6 | 4.1 | 5.5 | 5.0 | 6.3 | 7.6 | 8.1 | 10.8 | 12.1 | 6.3 | 9.4 | 28.1 | 6.7 | 1.6 | 7.0 | 2.6 | 7.8 | 9.0 | 6.6 | 4.7 |
| da | 2.0 | 8.9 | 4.0 | 5.2 | | 14.0 | 9.0 | 32.9 | 6.7 | 16.7 | 16.7 | 12.8 | 7.3 | 3.5 | 4.4 | 4.7 | 10.8 | 13.4 | 4.7 | 6.0 | 6.2 | 33.1 | 12.4 | 8.5 | 7.8 | 3.1 | 5.2 | 1.4 | 25.8 | 2.7 | 6.3 | 11.2 | 7.3 | 4.9 | | | |
| de | 2.4 | 9.7 | 4.9 | 8.1 | 16.9 | | 7.8 | 24.5 | 15.9 | 17.4 | 18.3 | 14.7 | 6.8 | 4.3 | 5.5 | 7.2 | 8.6 | 13.5 | 6.4 | 6.6 | 5.6 | 11.5 | 17.6 | 6.8 | 14.2 | 15.2 | 8.7 | 9.2 | 5.4 | 8.6 | 1.5 | 12.7 | 3.6 | 7.8 | 11.3 | 9.2 | 4.3 |
| el | 4.1 | 11.2 | 4.8 | 5.5 | 9.7 | 6.7 | | 27.9 | 8.0 | 18.8 | 16.3 | 13.5 | 10.1 | 4.0 | 5.6 | 5.3 | 13.1 | 15.3 | 5.5 | 6.1 | 10.2 | 9.3 | 8.7 | 5.1 | 18.0 | 18.3 | 10.4 | 8.4 | 3.1 | 6.4 | 2.0 | 7.2 | 3.4 | 7.2 | 14.4 | 8.7 | 5.3 |
| en | 11.9 | 23.9 | 14.7 | 15.5 | 30.9 | 20.4 | 27.1 | | 22.6 | 35.8 | 32.6 | 25.1 | 24.3 | 17.3 | 18.8 | 13.5 | 28.8 | 29.5 | 10.2 | 18.6 | 21.8 | 31.8 | 25.1 | 12.0 | 31.4 | 37.0 | 20.4 | 17.4 | 13.8 | 16.5 | 5.5 | 29.1 | 10.3 | 17.6 | 26.9 | 18.0 | 10.7 |
| eo | 1.8 | 6.4 | | 7.3 | 7.4 | 13.5 | 8.1 | 23.1 | | 16.1 | 17.6 | 12.7 | 10.5 | 1.9 | 3.0 | 4.8 | 9.2 | 13.9 | 2.5 | 4.8 | 6.2 | 7.2 | 12.1 | 6.7 | 12.4 | 16.0 | 6.9 | 8.6 | 7.4 | 5.6 | 0.6 | 8.8 | 1.8 | 5.4 | 7.7 | 5.2 | 3.6 |
| es | 6.2 | 14.3 | 6.8 | 8.0 | 15.7 | 12.9 | 16.4 | 33.2 | 13.5 | | 25.6 | 19.9 | 30.1 | 8.1 | 8.7 | 7.7 | 16.1 | 23.8 | 7.9 | 9.9 | 11.9 | 13.3 | 14.1 | 8.1 | 27.6 | 27.8 | 14.7 | 11.6 | 6.0 | 8.1 | 2.6 | 13.7 | 5.2 | 10.0 | 17.8 | 12.3 | 6.6 |
| fr | 6.0 | 12.7 | 5.0 | 8.3 | 15.6 | 14.5 | 15.4 | 31.6 | 18.6 | 26.4 | | 17.2 | 6.8 | 6.9 | 7.7 | 15.0 | 24.6 | 7.3 | 8.7 | 10.7 | 13.1 | 15.9 | 4.3 | 29.0 | 17.0 | 11.1 | 6.6 | | | | | | | | | | |
| fr-ca | 4.9 | 12.5 | 2.8 | 7.8 | 14.3 | 12.9 | 15.4 | 27.8 | 14.0 | 23.7 | | 18.1 | 6.8 | 6.8 | 7.9 | 13.7 | 23.4 | 7.5 | 8.8 | 10.3 | | 15.1 | 7.2 | 18.6 | 23.2 | 15.3 | 11.3 | 6.1 | 6.6 | 3.3 | 12.5 | 5.0 | 9.7 | 15.4 | | | 6.2 |
| gl | 2.6 | 7.3 | 4.7 | 2.9 | 7.4 | 5.3 | 8.5 | 23.4 | 9.2 | 34.4 | 16.0 | 15.2 | | 2.5 | 3.5 | 2.8 | 9.8 | 19.3 | 3.6 | 4.2 | 5.6 | 6.9 | 6.5 | 4.3 | 22.4 | 23.7 | 7.7 | 5.9 | 1.7 | 3.1 | 0.2 | 4.3 | 2.3 | 3.9 | 9.4 | 5.8 | 4.2 |
| he | 3.4 | 5.7 | 1.8 | 3.7 | 6.1 | 5.4 | 6.3 | 25.7 | 4.2 | 15.3 | 13.6 | 10.3 | 5.1 | | 2.5 | 3.2 | 8.9 | 11.2 | 4.1 | 4.5 | 4.6 | 5.4 | 6.5 | 3.7 | 13.7 | 15.0 | 6.3 | 8.9 | 1.9 | 2.6 | 1.0 | 5.2 | 1.8 | 5.1 | 10.7 | 6.9 | 4.6 |
| hr | 1.6 | 8.7 | 29.9 | 7.0 | 6.5 | 5.8 | 6.6 | 24.4 | 4.4 | 12.6 | 9.9 | 7.8 | 5.7 | 1.5 | | 4.3 | 8.2 | 10.4 | 1.8 | 4.0 | 14.1 | 5.3 | 6.1 | 5.4 | 12.1 | 12.6 | 7.3 | 8.3 | 4.6 | 12.6 | 11.7 | 6.0 | 1.9 | 6.9 | 9.4 | 4.8 | 2.9 |
| hu | 1.6 | 5.6 | 2.6 | | 5.9 | 7.0 | 5.5 | 16.7 | 6.5 | 10.8 | 10.9 | 9.3 | 3.9 | 2.1 | 3.6 | | 6.6 | 8.2 | 4.4 | 6.2 | 3.9 | 4.5 | 6.0 | 4.3 | 9.7 | 9.9 | 7.1 | 5.8 | 3.4 | 4.2 | 1.2 | 5.3 | 3.0 | 4.4 | 8.5 | 6.5 | 4.2 |
| id | 4.1 | 9.1 | 4.2 | 5.2 | 10.1 | 6.7 | 11.2 | 24.9 | 8.2 | 16.4 | 15.1 | 11.1 | 9.9 | 5.1 | 5.6 | 5.2 | | 12.7 | 5.8 | 9.1 | 7.0 | 10.0 | 9.4 | 5.5 | 14.6 | 14.9 | 6.6 | 4.8 | 1.4 | 7.3 | 18.5 | 11.0 | 6.2 | | | | |
| it | 5.3 | 11.7 | 5.0 | 6.9 | 13.1 | 11.5 | 14.5 | 30.0 | 13.9 | 26.4 | 24.9 | 20.0 | 19.3 | 6.2 | 7.0 | 6.7 | 14.0 | | 7.3 | 9.0 | 9.9 | 13.3 | 12.8 | 7.3 | 22.8 | 24.9 | 13.3 | 10.2 | 5.4 | 7.1 | 2.3 | 11.9 | 4.6 | 8.8 | 15.3 | 10.7 | 5.8 |
| ja | 1.4 | 1.9 | 0.7 | 1.8 | 3.1 | 2.7 | 2.5 | 7.9 | 2.2 | 6.0 | 6.0 | 4.7 | 2.3 | 1.4 | 1.3 | 2.0 | 3.5 | 4.6 | | 16.9 | 1.6 | 2.6 | 3.1 | 1.8 | 5.1 | 4.9 | 2.8 | 2.7 | 1.2 | 1.8 | 0.5 | 2.6 | 1.9 | 2.2 | 5.9 | | |
| ko | 0.9 | 1.7 | | 1.3 | 2.0 | 1.7 | 1.7 | 8.7 | 1.3 | 4.7 | 4.4 | 3.4 | 1.5 | 0.9 | 0.7 | 1.5 | 3.1 | 3.2 | 9.2 | | 1.2 | 1.3 | 1.9 | 1.4 | 3.5 | 3.3 | 2.0 | 2.1 | 1.0 | 1.5 | 0.4 | 1.5 | 1.4 | 1.5 | 4.3 | | |
| mk | 2.4 | 18.2 | 12.0 | 5.4 | 7.2 | 4.3 | 10.3 | 23.4 | 8.9 | 15.2 | 11.5 | 10.0 | 5.4 | 4.0 | 12.6 | 3.7 | 8.9 | 11.3 | 3.5 | 4.7 | | 7.5 | 6.7 | 4.5 | 13.9 | 15.6 | 8.3 | 7.5 | 3.5 | 7.1 | 3.7 | 5.6 | 2.0 | 6.4 | 10.9 | 6.3 | 4.3 |
| nb | 2.7 | 8.5 | | 5.4 | 32.7 | 9.8 | 8.9 | 35.1 | 9.5 | 17.0 | 14.6 | | 8.0 | 3.1 | 3.9 | 4.5 | 9.9 | 15.0 | 5.5 | 5.7 | 6.7 | | 4.7 | 14.2 | 9.5 | 7.5 | 3.2 | 3.9 | 0.6 | 26.3 | 1.9 | 6.2 | 7.8 | | | | |
| nl | 2.4 | 8.2 | 2.9 | 5.9 | 14.2 | 16.1 | 8.4 | 26.5 | 13.4 | 16.8 | 16.7 | 13.5 | 7.3 | 3.9 | 4.8 | 5.3 | 11.4 | 13.3 | 5.2 | 5.9 | 5.9 | | 5.3 | 13.8 | 15.4 | 7.8 | 7.6 | 4.1 | 5.1 | 1.6 | 11.1 | 3.2 | 6.1 | 10.6 | 8.0 | 5.1 |
| pl | 1.8 | 7.4 | 2.9 | 8.2 | 6.6 | 6.5 | 5.4 | 15.1 | 7.5 | 11.4 | 11.3 | 8.6 | 5.2 | 2.3 | 4.8 | 4.0 | 7.5 | 8.6 | 4.2 | 5.7 | 5.2 | 4.1 | 6.2 | | 9.6 | 9.9 | 6.2 | 9.5 | 6.2 | 5.3 | 1.5 | 5.6 | 2.4 | 8.9 | 7.7 | 6.3 | 3.6 |
| pt | 7.4 | 15.2 | 7.0 | 8.0 | 15.5 | 13.2 | 18.7 | 35.0 | 12.0 | 32.4 | 26.7 | 19.0 | 23.0 | 8.8 | 10.3 | 8.4 | 17.5 | 24.4 | 7.8 | 10.6 | 13.1 | 14.3 | 14.9 | 8.9 | | 15.3 | 11.7 | 6.6 | 8.1 | 2.0 | 14.3 | 6.2 | 9.3 | 18.0 | 12.9 | 5.8 |
| pt-br | 6.5 | 14.7 | 7.4 | 8.6 | 16.8 | 12.9 | 17.6 | 37.3 | 16.0 | 31.0 | 26.6 | 20.3 | 23.0 | 8.7 | 9.8 | 8.1 | 18.6 | 24.8 | 7.8 | 10.7 | 12.5 | 14.8 | 8.5 | | 15.1 | 11.8 | 6.4 | 8.9 | 2.8 | 14.6 | 5.3 | 10.8 | 18.8 | 13.2 | 6.7 | |
| ro | 3.2 | 9.7 | 3.7 | 5.1 | 9.4 | 7.5 | 10.4 | 25.0 | 6.7 | 18.8 | 19.3 | 14.6 | 10.0 | 4.0 | 5.7 | 5.5 | 11.0 | 15.5 | 4.3 | 6.4 | 7.3 | 8.0 | 8.1 | 5.2 | 15.4 | 17.7 | | 8.0 | 3.6 | 5.0 | 1.9 | 7.0 | 3.3 | 6.6 | 12.7 | 7.8 | 4.9 |
| ru | 3.3 | 12.6 | 4.2 | 7.7 | 8.5 | 8.8 | 8.3 | 18.7 | 9.9 | 14.3 | 14.5 | 11.0 | 6.0 | 4.9 | 6.8 | 5.6 | 9.5 | 11.7 | 6.1 | 7.7 | 7.4 | 8.0 | 8.1 | 8.9 | 12.4 | 13.9 | 8.2 | | 5.8 | 5.4 | 2.7 | 8.2 | 2.9 | 22.5 | 11.5 | 9.1 | 5.2 |
| sk | 0.7 | 5.1 | 2.7 | 27.0 | 4.3 | 5.7 | 3.2 | 16.9 | 9.3 | 9.4 | 8.5 | 6.7 | 2.7 | 1.0 | 5.1 | 3.7 | 4.9 | 5.0 | 2.3 | 3.9 | 3.5 | 4.5 | 5.0 | 7.1 | 7.8 | 8.5 | 3.5 | 6.6 | | 5.0 | 1.5 | 4.3 | 1.6 | 5.4 | 5.4 | 2.3 | 2.5 |
| sl | 1.2 | 6.2 | 7.6 | 5.5 | 4.7 | 7.6 | 5.8 | 17.3 | 5.9 | 11.4 | 8.5 | 6.4 | 3.2 | 1.2 | 11.2 | 3.9 | 6.5 | 7.8 | 2.7 | 4.2 | 6.3 | 4.3 | 5.5 | 4.8 | 9.9 | 4.8 | 5.9 | 3.6 | | 1.9 | 4.1 | 2.2 | 4.3 | 7.4 | 3.8 | 2.6 | |
| sr | 1.8 | 7.6 | 33.2 | 5.1 | 3.7 | 3.8 | 5.8 | 22.8 | 3.2 | 11.9 | 9.1 | 7.5 | 3.5 | 1.1 | 30.4 | 2.7 | 6.1 | 8.2 | 1.2 | 2.9 | 13.7 | 3.3 | 3.3 | 3.9 | 10.8 | 11.3 | 5.6 | 7.2 | 2.8 | 9.5 | | 3.0 | 1.1 | 5.7 | 6.8 | 4.3 | 3.1 |
| sv | 2.2 | 7.4 | 4.8 | 5.9 | 26.5 | 12.6 | 8.1 | 31.8 | 11.0 | 16.9 | 15.7 | 10.7 | 7.1 | 3.3 | 5.5 | 5.4 | 11.4 | 16.3 | 4.8 | 5.2 | 25.4 | 11.5 | 5.8 | 15.0 | 17.4 | 7.9 | 8.1 | 3.8 | 4.7 | 1.0 | | 3.2 | 6.9 | 12.9 | 7.8 | 4.8 | |
| tr | 2.2 | 3.5 | 2.0 | 2.6 | 3.9 | 4.1 | 4.7 | 15.9 | 2.9 | 9.4 | 7.7 | 6.1 | 3.6 | 1.6 | 2.1 | 3.4 | 6.7 | 6.4 | 4.3 | 7.0 | 3.5 | 3.1 | 4.2 | 2.5 | 9.0 | 8.4 | 4.6 | 4.0 | 1.8 | 2.3 | 0.8 | 3.5 | | 3.3 | 8.2 | 6.7 | 4.4 |
| uk | 2.9 | 12.3 | 5.3 | 7.4 | 7.5 | 7.5 | 8.4 | 20.7 | 6.5 | 14.2 | 14.1 | 11.2 | 5.5 | 3.5 | 6.6 | 4.7 | 9.5 | 11.2 | 4.9 | 5.8 | 7.2 | 6.3 | 6.9 | 9.6 | | 12.9 | 7.2 | 23.5 | 4.9 | 5.7 | 2.6 | 6.9 | 2.6 | | 11.4 | 7.9 | 4.9 |
| vi | 4.2 | 7.5 | 4.0 | 4.7 | 8.5 | 6.0 | 8.8 | 20.2 | 7.3 | 13.7 | 13.2 | 9.9 | 6.5 | 4.6 | 4.9 | 4.7 | 14.7 | 10.7 | 5.6 | 9.3 | 6.9 | 5.7 | 7.3 | 4.5 | 13.0 | 14.1 | 8.5 | 7.2 | 3.4 | 4.6 | 1.7 | 8.2 | 4.0 | 6.7 | | 9.9 | 6.7 |
| zh-cn | 2.1 | 3.2 | 1.0 | 2.2 | 3.8 | 3.2 | 4.5 | 11.8 | 3.8 | 8.2 | 7.6 | | 3.2 | 1.7 | 1.9 | 3.0 | 6.6 | 6.0 | | 3.4 | | 3.8 | 2.2 | 7.1 | 7.9 | 4.1 | 4.1 | 1.6 | 2.4 | 0.9 | 3.1 | 2.3 | 3.0 | 10.8 | | | |
| zh-tw | 2.2 | 3.1 | 1.1 | 2.1 | 3.7 | 2.8 | 3.9 | 10.7 | 3.4 | 7.5 | 7.2 | 6.1 | 2.8 | 1.8 | 1.6 | 3.0 | 6.2 | 5.4 | | 2.8 | | 3.5 | 2.3 | 6.3 | 6.9 | 3.5 | 3.9 | 1.4 | 2.1 | 0.9 | 3.0 | 2.4 | 2.9 | 10.0 | | | |

Table 5: BLEU scores on the TED test set as proposed in (Qi et al., 2018). NMT systems were trained on bitexts mined in Wikipedia only (with at least twenty-five thousand parallel sentences). No other resources were used.

usually not the case when translating between other major languages, for example in Asia. Let us list some interesting language pairs for which we were able to mine more than hundred thousand sentences: Korean/Japanese (222k), Russian/Japanese (196k), Indonesian/Vietnamese (146k), or Hebrew/Romance languages (120–150k sentences).

Overall, we were able to extract at least ten thousand parallel sentences for 85 different languages.[13] For several low-resource languages, we were able to extract more parallel sentences with other languages than English. These include, among others, Aragonse with Spanish, Lombard with Italian, Breton with several Romance languages, Western Frisian with Dutch, Luxembourgish with German or Egyptian Arabic and Wu Chinese with the respective major language.

Finally, Cebuano (ceb) falls clearly apart: it has a rather huge Wikipedia (17.9M filtered sentence), but most of it was generated by a bot, as for the Waray language[14]. This certainly explains that only a very small number of parallel sentences could be extracted. Although the same bot was also used to generate articles in the Swedish Wikipedia, our alignments seem to be better for that language.

## 5.2 QUALITATIVE EVALUATION

Aiming to perform a large-scale assessment of the quality of the extracted parallel sentences, we trained NMT systems on the extracted parallel sentences. We identified a publicly available data set which provide test sets for many language pairs: translations of TED talks as proposed in the context of a study on pretrained word embeddings for NMT[15] (Qi et al., 2018). We would like to emphasize that we did not use the training data provided by TED – we only trained on the mined sentences from Wikipedia. The goal of this study is not to build state-of-the-art NMT system for for the TED task, but to get an estimate of the quality of our extracted data, for many language pairs. In

---

[13]99 languages have more than 5,000 parallel sentences.

[14]https://en.wikipedia.org/wiki/Lsjbot

[15]https://github.com/neulab/word-embeddings-for-nmt

particular, there may be a mismatch in the topic and language style between Wikipedia texts and the transcribed and translated TED talks.

For training NMT systems, we used a transformer model from `fairseq` (Ott et al., 2019) with the parameter settings shown in Figure 2 in the appendix. For preprocessing, the text was tokenized using the Moses tokenizer (without true casing) and a 5000 subword vocabulary was learnt using SentencePiece (Kudo & Richardson, 2018). Decoding was done with beam size 5 and length normalization 1.2.

We evaluate the trained translation systems on the TED dataset (Qi et al., 2018). The TED data consists of parallel TED talk transcripts in multiple languages, and it provides development and test sets for 50 languages. Since the development and test sets were already tokenized, we first detokenize them using Moses. We trained NMT systems for all possible language pairs with more than twenty-five thousand mined sentences. This gives us in total 1886 language pairs in 45 languages. We train $L_1 \to L_2$ and $L_2 \to L_1$ with the same mined bitexts $L_1/L_2$. Scores on the test sets were computed with SacreBLEU (Post, 2018). Table 5 summarizes all the results. Due to space constraints, we are unable to report BLEU score for all language combinations in that table. Some additional results are reported in Table 7 in the annex. 23 NMT systems achieve BLEU scores over 30, the best one being 37.3 for Brazilian Portuguese to English. Several results are worth mentioning, like Farsi/English: 16.7, Hebrew/English: 25.7, Indonesian/English: 24.9 or English/Hindi: 25.7 We also achieve interesting results for translation between various non English language pairs for which it is usually not easy to find parallel data, e.g. Norwegian $\leftrightarrow$ Danish $\approx$33, Norwegian $\leftrightarrow$ Swedish $\approx$25, Indonesian $\leftrightarrow$ Vietnamese $\approx$16 or Japanese / Korean $\approx$17.

Our results on the TED set give an indication on the quality of the mined parallel sentences. These BLEU scores should be of course appreciated in context of the sizes of the mined corpora as given in Table 4. Obviously, we can not exclude that the provided data contains some wrong alignments even though the margin is large. Finally, we would like to point out that we run our approach on all available languages in Wikipedia, independently of the quality of LASER's sentence embeddings for each one.

# 6 CONCLUSION

We have presented an approach to systematically mine for parallel sentences in the textual content of Wikipedia, for all possible language pairs. We use a recently proposed mining approach based on massively multilingual sentence embeddings (Artetxe & Schwenk, 2018a) and a margin criterion (Artetxe & Schwenk, 2018b). The same approach is used for all language pairs without the need of a language specific optimization. In total, we make available 135M parallel sentences in 85 languages, out of which only 34M sentences are aligned with English. We were able to mine more than ten thousands sentences for 1620 different language pairs. This corpus of parallel sentences is freely available.[16] We also performed a large scale evaluation of the quality of the mined sentences by training 1886 NMT systems and evaluating them on the 45 languages of the TED corpus (Qi et al., 2018).

This work opens several directions for future research. The mined texts could be used to first retrain LASER's multilingual sentence embeddings with the hope to improve the performance on low-resource languages, and then to rerun mining in Wikipedia. This process could be iteratively repeated. We also plan to apply the same methodology to other large multilingual collections. The monolingual texts made available by ParaCrawl or CommonCrawl[17] are good candidates.

We expect that the WikiMatrix corpus has mostly well-formed sentences and it should not contain social media language. The mined parallel sentences are not limited to specific topics like many of the currently available resources (parliament proceedings, subtitles, software documentation, ...), but are expected to cover many topics of Wikipedia. The fraction of unedited machine translated text is also expected to be low. We hope that this resource will be useful to support research in multilinguality, in particular machine translation.

---

[16]Anonymized for review

[17]http://commoncrawl.org/

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

# A APPENDIX

Table 6 provides the amounts of mined parallel sentences for languages which have a rather small Wikipedia. Aligning those languages obviously yields to a very small amount of parallel sentences. Therefore, we only provide these results for alignment with high resource languages. It is also likely that several of these alignments are of low quality since the LASER embeddings were not directly trained on most these languages, but we still hope to achieve reasonable results since other languages of the same family may be covered.

| ISO | Name | Language Family | size | ca | da | de | en | es | fr | it | nl | pl | pt | sv | ru | zh | total |
|---|---|---|---|---|---|---|---|---|---|---|---|---|---|---|---|---|---|
| an | Aragonese | Romance | 222 | 24 | 7 | 12 | 23 | 33 | 16 | 13 | 9 | 10 | 14 | 9 | 11 | 6 | 324 |
| arz | Egyptian Arabic | Arabic | 120 | 7 | 6 | 11 | 18 | 12 | 12 | 10 | 8 | 9 | 10 | 8 | 12 | 7 | 278 |
| as | Assamese | Indo-Aryan | 124 | 8 | 6 | 11 | 7 | 11 | 12 | 10 | 9 | 9 | 8 | 8 | 9 | 3 | 216 |
| azb | South Azerbaijani | Turkic | 398 | 6 | 4 | 9 | 8 | 9 | 10 | 9 | 7 | 6 | 8 | 6 | 7 | 3 | 172 |
| bar | Bavarian | Germanic | 214 | 7 | 6 | 41 | 16 | 12 | 12 | 10 | 8 | 9 | 10 | 8 | 10 | 5 | 261 |
| bpy | Bishnupriya | Indo-Aryan | 128 | 2 | 1 | 4 | 4 | 3 | 4 | 2 | 2 | 3 | 2 | 2 | 3 | 1 | 71 |
| br | Breton | Celtic | 413 | | | 20 | 16 | 22 | 23 | 22 | | | 19 | | 16 | 6 | 200 |
| ce | Chechen | Northeast Caucasian | 315 | 2 | 1 | 2 | 2 | 2 | 2 | 2 | 2 | 2 | 2 | 2 | 2 | 1 | 56 |
| ceb | Cebuano | Malayo-Polynesian | 17919 | 14 | 9 | 22 | 29 | 27 | 24 | 24 | 15 | 17 | 20 | 55 | 21 | 9 | 594 |
| ckb | Central Kurdish | Iranian | 127 | 2 | 2 | 6 | 8 | 5 | 5 | 4 | 4 | 4 | 4 | 3 | 6 | 4 | 113 |
| cv | Chuvash | Turkic | 198 | 4 | 3 | 5 | 4 | 6 | 6 | 7 | 5 | 4 | 6 | 5 | 8 | 2 | 129 |
| dv | Maldivian | Indo-Aryan | 52 | 2 | 2 | 5 | 6 | 4 | 4 | 3 | 3 | 3 | 3 | 3 | 5 | 3 | 96 |
| fo | Faroese | Germanic | 114 | 13 | 12 | 14 | 32 | 21 | 18 | 15 | 11 | 11 | 17 | 12 | 13 | 6 | 335 |
| fy | Western Frisian | Germanic | 493 | 13 | 8 | 16 | 32 | 21 | 18 | 17 | 38 | 12 | 18 | 13 | 14 | 5 | 453 |
| gd | Gaelic | Celtic | 66 | 1 | 1 | 1 | 1 | 1 | 1 | 1 | 1 | 1 | 1 | 1 | 1 | 1 | 41 |
| ga | Irish | Irish | 216 | | 2 | 3 | 4 | 3 | 3 | 3 | 2 | 2 | 3 | 2 | 3 | 1 | 70 |
| gom | Goan Konkami | Indo-Aryan | 69 | 9 | 7 | 10 | 8 | 13 | 13 | 13 | 9 | 9 | 11 | 9 | 10 | 4 | 240 |
| ht | Haitian Creole | Creole | 60 | 2 | 1 | 3 | 4 | 3 | 4 | 3 | 2 | 3 | 2 | 2 | 3 | 1 | 72 |
| ilo | Iloko | Philippine | 63 | 3 | 2 | 4 | 5 | 4 | 4 | 4 | 3 | 3 | 4 | 3 | 4 | 2 | 96 |
| io | Ido | constructed | 153 | 5 | 3 | 6 | 11 | 7 | 7 | 5 | 5 | 6 | 5 | 5 | 5 | 3 | 143 |
| jv | Javanese | Malayo-Polynesian | 220 | 8 | 5 | 8 | 13 | 12 | 10 | 11 | 8 | 7 | 11 | 8 | 8 | 3 | 219 |
| ka | Georgian | Kartvelian | 480 | 11 | 7 | 15 | 12 | 16 | 17 | 16 | 12 | 11 | 14 | 12 | 13 | 5 | 288 |
| ku | Kurdish | Iranian | 165 | 5 | 4 | 8 | 5 | 8 | 7 | 8 | 7 | 6 | 7 | 6 | 6 | 3 | 222 |
| la | Latin | Romance | 558 | 12 | 9 | 17 | 32 | 20 | 18 | 17 | 12 | 13 | 18 | 13 | 14 | 6 | 478 |
| lb | Luxembourgish | Germanic | 372 | 12 | 7 | 26 | 22 | 19 | 18 | 15 | 11 | 11 | 16 | 12 | 11 | 4 | 305 |
| lmo | Lombard | Romance | 147 | 6 | 3 | 7 | 10 | 7 | 7 | 11 | 6 | 5 | 7 | 5 | 5 | 3 | 144 |
| mg | Malagasy | Malayo-Polynesian | 263 | 6 | 5 | 9 | 13 | 9 | 12 | 8 | 7 | 7 | 7 | 8 | 7 | 4 | 199 |
| mhr | Eastern Mari | Uralic | 61 | 3 | 2 | 4 | 3 | 4 | 4 | 5 | 3 | 3 | 4 | 3 | 4 | 2 | 96 |
| min | Minangkabau | Malayo-Polynesian | 255 | 4 | 2 | 6 | 7 | 5 | 5 | 5 | 4 | 4 | 4 | 5 | 5 | 2 | 121 |
| mn | Mongolian | Mongolic | 255 | 4 | 3 | 7 | 5 | 6 | 6 | 7 | 6 | 5 | 5 | 5 | 5 | 3 | 197 |
| mwl | Mirandese | Romance | 64 | 6 | 3 | 4 | 10 | 8 | 6 | 5 | 3 | 4 | 34 | 3 | 4 | 2 | 154 |
| nds nl | Low German/Saxon | Germanic | 65 | 5 | 4 | 6 | 10 | 7 | 7 | 6 | 15 | 5 | 6 | 5 | 5 | 3 | 151 |
| ps | Pashto | Iranian | 89 | | 2 | 3 | 2 | 3 | 3 | 3 | 3 | 3 | 3 | 3 | 3 | 1 | 73 |
| rm | Romansh | Italic | 57 | 2 | 2 | 10 | 5 | 4 | 4 | 3 | 2 | 3 | 3 | 3 | 3 | 1 | 86 |
| sah | Yakut | Turkic/Sib | 134 | 4 | 3 | 7 | 5 | 6 | 6 | 6 | 5 | 5 | 5 | 5 | 6 | 3 | 134 |
| scn | Sicilian | Romance | 81 | 5 | 3 | 6 | 9 | 7 | 7 | 11 | 5 | 5 | 6 | 5 | 5 | 2 | 143 |
| sd | Sindhi | Iranian | 115 | | 3 | 9 | | 8 | 8 | 7 | 7 | 6 | 7 | 5 | 8 | 5 | 152 |
| su | Sundanese | Malayo-Polynesian | 120 | 4 | 3 | 5 | 7 | 6 | 5 | 6 | 4 | 4 | 5 | 4 | 4 | 2 | 117 |
| tk | Turkmen | Turkic | 56 | 2 | 2 | 3 | 3 | 4 | 3 | 4 | 2 | 2 | 4 | 2 | 3 | 1 | 76 |
| tg | Tajik | Iranian | 248 | 5 | 4 | 11 | 15 | 9 | 9 | 8 | 8 | 7 | 8 | 6 | 10 | 6 | 192 |
| ug | Uighur | Turkic | 83 | 4 | 3 | 9 | 10 | 7 | 8 | 6 | 6 | 5 | 6 | 5 | 9 | 6 | 168 |
| ur | Urdu | Indo-Aryan | 150 | 2 | 2 | 3 | 5 | 3 | 3 | 3 | 3 | 3 | 3 | 3 | 3 | 2 | 123 |
| wa | Walloon | Romance | 56 | 3 | 2 | 4 | 5 | 5 | 4 | 4 | 3 | 3 | 4 | 3 | 3 | 2 | 93 |
| wuu | Wu Chinese | Chinese | 75 | 8 | 6 | 11 | 17 | 12 | 11 | 10 | 8 | 9 | 11 | 9 | 10 | 43 | 283 |
| yi | Yiddish | Germanic | 131 | 3 | 2 | 4 | 3 | 4 | 4 | 5 | 3 | 3 | 4 | 3 | 4 | 1 | 92 |

Table 6: WikiMatrix (part 2): number of extracted sentences (in thousands) for languages with a rather small Wikipedia. Alignments with other languages yield less than 5k sentences and are omitted for clairty.

Table 2 gives the detailed configuration which was used to train NMT models on the mined data in Section 5.

```
--arch transformer
--share-all-embeddings
--encoder-layers 5
--decoder-layers 5
--encoder-embed-dim 512
--decoder-embed-dim 512
--encoder-ffn-embed-dim 2048
--decoder-ffn-embed-dim 2048
--encoder-attention-heads 2
--decoder-attention-heads 2
--encoder-normalize-before
--decoder-normalize-before
--dropout 0.4
--attention-dropout 0.2
--relu-dropout 0.2
--weight-decay 0.0001
--label-smoothing 0.2
--criterion label_smoothed_cross_entropy
--optimizer adam
--adam-betas '(0.9, 0.98)'
--clip-norm 0
--lr-scheduler inverse_sqrt
--warmup-update 4000
--warmup-init-lr 1e-7
--lr 1e-3 --min-lr 1e-9
--max-tokens 4000
--update-freq 4
--max-epoch 100
--save-interval 10
```

Figure 2: Model settings for NMT training with `fairseq`

Finally, Table 7 gives the BLEU scores on the TED corpus when translating into and from English for some additional languages.

| Lang | xx → en | en → xx |
|------|---------|---------|
| et   | 15.9    | 14.3    |
| eu   | 10.1    | 7.6     |
| fa   | 16.7    | 8.8     |
| fi   | 10.9    | 10.9    |
| lt   | 13.7    | 10.0    |
| hi   | 17.8    | 21.9    |
| mr   | 2.6     | 3.5     |

Table 7: BLEU scores on the TED test set as proposed in (Qi et al., 2018). NMT systems were trained on bitexts mined in Wikipedia only. No other resources were used.

