# OpenReview forum: "WikiMatrix: Mining 135M Parallel Sentences in 1620 Language Pairs from Wikipedia"
_ICLR.cc/2020/Conference — Reject_

### Official Review · AnonReviewer1 · 2019-10-21
**Official Blind Review #1**

**Rating:** 6

**Review:**

The paper presents WikiMatrix, an approach to automatically extract parallel sentences from the free text content of Wikipedia. The paper considers 1620 languages and the final dataset contains 135M parallel sentences. The language pairs are general and therefore the data does not require the use of English as a common language between two other languages.

To evaluate the quality of the extracted pairs, a neural machine translation system has been trained on them and tested on the TED dataset, obtaining good results in terms of BLEU score.

The article provides information on the system used to extract parallel sentences and opens up different directions for future investigations.

The dataset seems, from the given details, useful. However, without access to the data and, more importantly, extensive testing of it, it is difficult to say how and how much it would help the advancement of the field. For the moment it seems to be good. However, I am not really sure that this paper could be of interest to a wide audience, except for those involved in machine translation.

In general, the article describes everything at a high level, without going into the real details.
An example of this is on page 6, section 4.2, where the article says that its purpose is to compare different mining parameters, but I do not see any real comparison. Some words are spent for the mining threshold, but there is no real comparison, while other possible parameters are not considered at all.

For this reason, I would tend to give a low score, which does not mean that the dataset is not good. It means that the real content of the paper seems to me to be too little to be published at ICLR, since the paper only informs about the presence of this new dataset, saying that it contains a large number of sentences and seems to allow good translations based on the results of a preliminary test.

Typos:
- on page 9 "Aragonse"
- on page 9, end penultimate line, the word "for" is repeated.

**Experience Assessment:**

I do not know much about this area.

**Review Assessment: Checking Correctness Of Derivations And Theory:**

N/A

**Review Assessment: Checking Correctness Of Experiments:**

N/A

**Review Assessment: Thoroughness In Paper Reading:**

N/A

---

> ### Author Response · Authors · 2019-11-15
> **We would like to thank the reviewer for his work. In the following, we will comment on the remarks of the reviewer:**
>
> 1) “However, without access to the data and, more importantly, extensive testing of it, it is difficult to say how and how much it would help the advancement of the field”
> We would like to point out that we have already open sourced the data. This is mentioned in the abstract and the conclusion of the paper (the download URL is not given to guarantee anonymity). In addition, we have trained more than 1800 neural machine translation systems covering 45 languages and report BLEU scores for all of them on the TED corpus. 23 systems achieve BLEU scores over 30. We believe that this qualifies for “extensive testing”, given that in most of the papers in NLP, evaluation is limited to a handful of tasks or language pairs.
>
> 2) “Everything is described at a high level without going into detail, e.g. no real comparison of the mining parameters.”
> We would like to point out that our algorithms are described in detail in section 4 which spans over 2 pages. In particular, our mining approach has only one parameter, the mining threshold. We provide an extensive study of the impact of this parameter in section 4.2 for four different language pairs (see Figure 1).
>
> 3) “Real content is too little to be published at ICLR, informs only on the presence of a new dataset”
> We would like to argue that this paper makes a substantial contribution for several reasons:
>  - this is the first approach which can be applied to many language pairs without the need to adapt it to the specific language pair. In fact, the corpus is particularly useful for research in low resource translation, which is an active research area.
> - the mined corpus is unique in its genre, with respect to the number of languages covered, including many low-resource languages, and the fact that we consider all possible language pairs (instead of the usual English/foreign pair).

---

> > ### Comment · AnonReviewer1 · 2019-11-15
> > **Response to authors**
> >
> > Regarding point 1, to maintain anonymity it was enough to create, for example, a temporary account on GitHub. However, I am aware of the problem of anonymity and my consideration was about a check that could be performed by the reviewers, not about the tests presented in the article.
> >
> > Regarding point 2, section 4 explains how the data corpus has been prepared and talks about threshold optimization. The sentence I mentioned about the comparison of different mining parameters at this point seems misleading in my opinion because there is only one parameter that is optimized (comparisons are implicit in optimization). Reading about the comparison of parameters I expected instead of a discussion about parameters and results obtained from different systems. There is instead a comparison between the application on different languages. Therefore, I thank you for the clarification that allowed me to better frame the article.
> >
> > Having said that, I would like to thank the authors for taking my comments well into consideration and for replying punctually to all of them, clarifying some of my doubts. I think I can happily increase my score.

---

### Official Review · AnonReviewer3 · 2019-10-24
**Official Blind Review #3**

**Rating:** 8

**Review:**

The paper creates a large dataset for machine translation, called WikiMatrix, that contains 135M parallel sentences in 1620 language pairs from Wikipedia. The paired sentences from different languages are mined based on the sentence embeddings. Training NMT systems based on the mined dataset, and comparing with those trained based on existing dataset, the authors claim that the quality of the dataset is good. The effect of thresholding values of similarity scores for selecting parallel sentences is studied. Since the data is huge, dimension reduction and data compression techniques are used for efficient mining. The study is the first one that systematically mine for parallel sentences of Wikipedia for a large number of languages. The results are solid and the dataset is valuable for research in multilinguality.

**Experience Assessment:**

I have read many papers in this area.

**Review Assessment: Checking Correctness Of Derivations And Theory:**

N/A

**Review Assessment: Checking Correctness Of Experiments:**

I assessed the sensibility of the experiments.

**Review Assessment: Thoroughness In Paper Reading:**

I read the paper at least twice and used my best judgement in assessing the paper.

---

> ### Author Response · Authors · 2019-11-15
> **We would like to thank the reviewer for his thorough review.**

---

### Official Review · AnonReviewer2 · 2019-10-24
**Official Blind Review #2**

**Rating:** 3

**Review:**

This ICLR submission deals with an strategy for the automatic extraction of parallel sentences from Wikipedia articles in 85 languages, based on multilingual sentence embeddings.
The review is delivered with the caveat that I am not an expert in this particulat field.
The paper is well written and structured, being within the scope of the conference.
The literature review is very up to date and overall relevant to provide an appropriate context to the investigation.
I reckon this is a very interesting piece of work, but also that it draws too heavily on previous work from which the study is just an incremntal extension.
Minor issues:
All acronyms in the text should be defined the first time they appear in the text.
1st sentence of section 2: typo on “comparable coprora”.

**Experience Assessment:**

I do not know much about this area.

**Review Assessment: Checking Correctness Of Derivations And Theory:**

I assessed the sensibility of the derivations and theory.

**Review Assessment: Checking Correctness Of Experiments:**

I assessed the sensibility of the experiments.

**Review Assessment: Thoroughness In Paper Reading:**

I read the paper at least twice and used my best judgement in assessing the paper.

---

> ### Author Response · Authors · 2019-11-15
> **We would like to thank the reviewer for his work. Please find below our comments.**
>
> We understood that the major concern of the reviewer is that this work is only an incremental extension of previous work. There is indeed a large body of research on bitext mining, as described in the related work section, which was recognized by the reviewer as very up-to-date.
>
> We would like to point out that, to the best of our knowledge, our work is the first one to systematically mine for parallel data in Wikipedia, with one unified approach. None of the preceding approaches could be applied to such a large number of languages (we handle 85 languages). Almost all of the existing approaches focus on alignment with English only, while we provide alignments for all 1620 language pairs (Table 4 of the paper).
>
> Finally, we would like to mention that we make freely available all the mined parallel sentences to foster research on multilingual models.
>
> We will change the text as requested, in particular the definition of the acronyms.

---

### Official Review · AnonReviewer4 · 2019-11-20
**Official Blind Review #4**

**Rating:** 3

**Review:**

The paper presents a multi-lingual multi-way pseudo-parallel text corpus automatically extracted from Wikipedia.

The authors use a variety of pre-existing techniques applied at large scale with substantial engineering effort to extract a large number of sentence pairs in 1620 language pairs from 85 languages.

In the proposed method 1) raw sentences are extracted from a Wikipedia dump, 2) LASER sentence embeddings and language IDs are computed for each sentence, 3) for each language pair candidate sentence pairs are extracted using a FAISS approximate K-nearest neighbor index on the cosine distance between sentence embeddings, 4) sentence similarity scores are computed between the candidate pairs using the "max margin" criterion of Artetxe & Schwenk, 2018 and finally 5) sentence pairs are selected according to a language-pair-agnostic threshold on the similarity scores.

The extraction method is symmetric w.r.t. language directions for each language pair.

Structural metadata of Wikipedia, such as cross-lingual document alignments, is deliberately not exploited (some discussion is provided but I would have preferred an empirical comparison of local vs global extraction).

The similarity threshold is determined by evaluating training corpora extracted at different thresholds on a machine translation task on De->En, De->Fr, Cs->De and Cs->Fr translation directions, evaluated on WMT newstest2014, and manually selecting the threshold based on BLEU scores. The paper also reports that combining the automatically extracted corpora with Europarl results in strong BLEU improvements over training only on Europarl. BLEU scores on TED test sets obtained using only the automatically extracted corpus are also reported. The corpus has been released.

Overall the methodology presented in the paper is strong and the corpus is likely going to become a valuable tool to build machine translation systems and other multi-lingual applications. However, I am concerned that ICLR 2020 may not be the appropriate venue for this paper, as in my understanding dataset release papers are not explicitly solicited in the Call for Papers https://iclr.cc/Conferences/2020/CallForPapers . The corpus generation method is based on existing techniques, and to the extent that the engineering effort is innovative, it might not necessarily transfer well to data sources other than Wikipedia, thus limiting its broad scientific value. Therefore I suggest to submit the paper to a different venue.



**Experience Assessment:**

I have published in this field for several years.

**Review Assessment: Checking Correctness Of Derivations And Theory:**

N/A

**Review Assessment: Checking Correctness Of Experiments:**

I assessed the sensibility of the experiments.

**Review Assessment: Thoroughness In Paper Reading:**

I read the paper thoroughly.

---

### Decision · Program_Chairs · 2019-12-19

**Decision:**

Reject

**Comment:**

The authors present an approach to large scale bitext extraction from Wikipedia. This builds heavily on previous work, with the novelty being somewhat minor efficient approximate K-nearest neighbor search and language agnostic parameters such as cutoffs. These techniques have not been validated on other data sets and it is unclear how well they generalise. The major contribution of the paper is the corpus created, consisting of 85 languages, 1620 language pairs and 135M parallel sentences, of which most do not include English. This corpus is very valuable and already in use in the field, but IMO ICLR is not the right venue for this kind of publication. There were four reviews, all broadly in agreement, and some discussion with the authors.